# Secure Physical Layer Network Coding versus Secure Network Coding [note 1]

**DOI:** 10.3390/e24010047

**Published:** 2021-12-27

**Authors:** Masahito Hayashi

**Affiliations:** 1Shenzhen Institute for Quantum Science and Engineering, Southern University of Science and Technology, Nanshan District, Shenzhen 518055, China; hayashi@sustech.edu.cn; 2International Quantum Academy (SIQA), Futian District, Shenzhen 518048, China; 3Guangdong Provincial Key Laboratory of Quantum Science and Engineering, Southern University of Science and Technology, Nanshan District, Shenzhen 518055, China; 4Graduate School of Mathematics, Nagoya University, Furocho, Chikusa-ku, Nagoya 464-8602, Japan

**Keywords:** secrecy analysis, secure communication, untrusted relay, network coding, physical layer security, cross-layer protocol

## Abstract

When a network has relay nodes, there is a risk that a part of the information is leaked to an untrusted relay. Secure network coding (secure NC) is known as a method to resolve this problem, which enables the secrecy of the message when the message is transmitted over a noiseless network and a part of the edges or a part of the intermediate (untrusted) nodes are eavesdropped. If the channels on the network are noisy, the error correction is applied to noisy channels before the application of secure NC on an upper layer. In contrast, secure physical layer network coding (secure PLNC) is a method to securely transmit a message by a combination of coding operation on nodes when the network is composed of set of noisy channels. Since secure NC is a protocol on an upper layer, secure PLNC can be considered as a cross-layer protocol. In this paper, we compare secure PLNC with a simple combination of secure NC and error correction over several typical network models studied in secure NC.

## 1. Introduction

Wireless communication networks with relay nodes have a risk for information leakage to untrusted relays. To resolve this problem, several studies [1,2,3,4,5,6] considered the relay terminals untrustworthy based on the result of secure computation-and-forward (CAF) tests [7,8,9,10,11], which is the main topic of the secure extension of physical layer network coding (PLNC), in short, secure PLNC. However, this type of security can be realized by the secure extension of network coding (NC), in short, secure NC, which is an upper-layered protocol to securely transmit a message via a noiseless network when a part of the edges and/or a part of the intermediate (untrusted) nodes are eavesdropped [12,13,14,15,16,17]. Since a wireless channel is disturbed by noise, an error correction needs to be applied to the channel. Then, secure NC is applied to noiseless channels virtually implemented by an error correction. In other words, the error correction and secure NC are separately performed in the different layers under the above scenario. In contrast, since secure PLNC combines both parts, it can be considered as a cross-layer protocol. In order to clarify the advantage of this cross-layer protocol, it is needed to compare secure PLNC with a simple combination of secure NC and error correction over wireless channels, and this comparison has not been studied yet. That is, this type of comparison is strongly required in the viewpoint of wireless communication networks.

Secure PLNC is based on PLNC [18,19,20], which efficiently transmits the modulo sum of two transmitters’ messages via a Gaussian channel. To guarantee the security, the preceding studies [7,8,9,10,11,21] invented a secure extension of PLNC, i.e., secure PLNC, which is a scheme to securely transmit a message by a combination of coding operations on nodes when the network is given as a set of noisy channels. Secure PLNC can be classified into two types. In the first case, secure NC is applied to the noiseless CAF process realized by PLNC. This method can be considered as a simple combination of secure NC and PLNC. The other type is a direct method to realize security in the PLNC. The typical example is secure CAF. The code of the latter type cannot be made by such a simple combination. All existing studies [7,8,9,10,11,21] belong to the latter case and address only a two-hop relay scheme or its simple extension, the multi-hop relay scheme, which are based on secure CAF to securely transmit the modulo sum of two input message when the channel is a noisy multiple access channel (MAC). Indeed, a secure NC can guarantee the secrecy for the eavesdropper that eavesdrops the channels. Several typical secure NCs cannot guarantee the secrecy when one of the intermediate (untrusted) nodes is eavesdropped. In this way, secure PLNC has an advantage under attacks on intermediate (untrusted) nodes.

However, the network models studied in secure NC are more advanced and more complicated, and no study discussed secure PLNC over such typical network models in secure NC. That is, the network models studied in secure PLNC is too limited and too primitive in the comparison with typical network models in secure NC. In other words, no prior study investigated the application of secure PLNC to such typical network models. In order for secure PLNC to overcome secure NC, we need to demonstrate that secure PLNC can be used in more advanced network models. At least, it is needed to study secure PLNC over typical network models in secure NC.

Since no existing paper has made the comparison between secure PLNC and the simple combination of secure NC and error correction, this paper aims to make this type of comparison under typical network models in secure NC. That is, this paper is the first study for secure PLNC over typical network models in secure NC, the butterfly network model and the network model composed of three source nodes under certain assumptions for the attack. Unfortunately, secure PLNC has a completely different mathematical structure from the simple combination of secure NC and error correction. Hence, it is quite difficult to construct a general theory to compare them. Due to this reason, we address two typical network models in the area of secure NC, the butterfly network [22] and a network with three source nodes, which is a special network model studied in [23].Then, we make the above comparison numerically over these two networks. Indeed, many existing studies [7,8,9,10,11] for secure PLNC employed lattice codes. Only the reference [21] studied it with BPSK modulation. Notice that the QPSK modulation can be considered as twice the use of the BPSK modulation.

For PLNC, references [24,25,26,27] discussed CAF based on lattice codes. Indeed, $2^{n}$-phase shift keying (PSK) modulation works for practical systems such as conventional satellite communications with LDPC codes [28]. In addition, references [29,30] demonstrated the efficiency of the CAF scheme composed of binary LDPC codes under the BPSK modulation. Reference [31] compared the BPSK modulation and the method based on lattice codes for CAF. Hence, to adopt the existing communication system, we focus on the BPSK modulation.

Although, this paper is the journal version of the preceding conference paper [32], this paper is different from the conference version as follows. First, the conference version gave the secure NC protocol only when *q* is not a power of 2. This paper additionally gives the secure NC protocol when *q* is a power of 2 (not 2). This kind of extension enables us to consider the new protocol given in Section 3.3.2. Second, the conference version discussed only one type of secure NC protocol. This paper additionally considers another type of protocol in secure PLNC (See Section 3.3.2 and Section 4.2.2). Totally, this paper discusses two types of protocols in secure PLNC. This additional protocol clarifies the merit of use of CAF. Third, the conference version compared the number of times slots only for two cases: secure NC without Gaussian MAC and secure PLNC with Gaussian MAC. Also, it did not considered the protocol in Section 3.3.2 and Section 4.2.2. This paper additionally considers another case: secure NC with Gaussian MAC. Further, this version discussed the transmission time by considering the information transmission rate when the asymptotically best code is employed. To make this additional comparison, analytical discussions are newly made in this version by using the mutual information. Also, this version newly contains numerical graphs (Figures 3 and 5) for this comparison. Due to this additional comparison, we can compare the transmission time.

The rest of this paper is organized as follows. First, Section 2 reviews the results in CAF and secure CAF, which is a typical example of secure PLNC. Next, Section 3 considers how secure communication can be implemented over the butterfly network based on secure PLNC. Finally, Section 4 discusses how secure communication can be implemented over a network with three source nodes based on secure PLNC. That is, Section 3 and Section 4 are devoted to our contribution.

## 2. CAF and Secure CAF

### 2.1. CAF

As the first step, we review existing results for secure CAF. For this aim, we prepare an important notation. The symbol ⊕ expresses the arithmetic sum over a finite field, and the symbol + denotes the sum over the real numbers. A typical setting for secure CAF has two transmitters, $V_{1}$ and $V_{2}$, and one receiver, *R*. Suppose that Transmitter $V_{i}$ has message $M_{i}\in F_{q}$, and Receiver *R* is linked by a (noisy) MAC that has two input variables from the two transmitters $V_{1}$ and $V_{2}$. In this scheme, Receiver *R* is required to obtain the modulo sum $M_{1}⊕M_{2}$ via the (noisy) MAC, as depicted in Figure 1.

Many papers proposed a protocol for CAF over a Gaussian MAC. Suppose that the transmitter $V_{i}$ sends the complex-valued variable $X_{i}$ for i=1,2. When the channel fading coefficients are given as $h_{1},h_{2}\in C$, Receiver *R* receives the complex-valued variable *Y* as:(1)$$\begin{array}{c} Y=h_{1}X_{1}+h_{2}X_{2}+N, \end{array}$$
where *N* is a complex Gaussian random variable with zero mean and a variance of one. The remaining part of this section assumes multiple uses of the above Gaussian MAC.

References [24,33,34] obtained an achievable rate under the energy constraint by using lattice codes. This rate is called the computation rate. Here, to seek a practical scheme, we consider the BPSK scheme, in which $X_{i}$ is coded to ${\left(-1\right)}^{A_{i}}$ with $A_{i}\in F_{2}$. Hence, (Equation 1) can be rewritten as:(2)$$\begin{array}{c} Y=h_{1}{\left(-1\right)}^{A_{1}}+h_{2}{\left(-1\right)}^{A_{2}}+N. \end{array}$$

The reference [35] showed that the rate $I{\left(Y;A_{1}⊕A_{2}\right)}_{\mathrm{Equation}(2)}$ is achieved when the task of CAF is imposed, where the mutual information is given by the independent and uniform random numbers $A_{1}$ and $A_{2}$. (More precisely, the quantity $I{\left(Y;A_{1}⊕A_{2}\right)}_{\mathrm{Equation}(2)}$ is defined as the mutual information when $A_{1}$ and $A_{2}$ are independently subject to the uniform distribution. This rule will be applied later when the equation number such as Equation (Equation 2) is given as a subscript of a mutual information). Then, references [29,30] studied LDPC codes, in particular, spatial coupling LDPC codes and regular LDPC codes, to achieve this task under the BPSK scheme. In fact, the method introduced by references [29,30] can be efficiently implemented with a rate close to $I{\left(Y;A_{1}⊕A_{2}\right)}_{\mathrm{Equation}(2)}$. Furthermore, the recent reference [36] studied its quantum extension.

### 2.2. Secure CAF

Next, we consider the secrecy condition for each message to Receiver *R* in addition to the correct decoding. This problem setting is called secure CAF. Here, Receiver *R* is required to obtain the modulo sum $M_{1}⊕M_{2}$ while the variable *Y* in Receiver *R*’s hand is required to be independent of $M_{1}$ and $M_{2}$. References [7,8,9,10,11] proposed an approach using lattice codes. Using an efficiently implementable algebraic for CAF given in [29,30], the reference [21] proposed an efficiently implementable code for secure CAF. (Here, a code is called an algebraic code when the encoding map preserves algebraic operation. For example, Reed Solomon codes and LDPC codes are algebraic codes.) It also showed that the rate $2I{\left(Y;A_{1}⊕A_{2}\right)}_{\mathrm{Equation}(2)}-I{\left(Y;A_{1},A_{2}\right)}_{\mathrm{Equation}(2)}$ is achievable in the BPSK scheme ([21], (29)), where the mutual information is given with the independent and uniform random numbers $A_{1}$ and $A_{2}$. That is, when the channel (Equation 2) is prepared and Receiver *R* colludes with no transmitter, secure CAF guarantees no information leakage of each message to Receiver *R* while Receiver *R* can recover the sum $M_{1}⊕M_{2}$. In the code in [21], $I{\left(Y;A_{1}⊕A_{2}\right)}_{\mathrm{Equation}(2)}$ is the rate of CAF, and $I{\left(Y;A_{1},A_{2}\right)}_{\mathrm{Equation}(2)}-I{\left(Y;A_{1}⊕A_{2}\right)}_{\mathrm{Equation}(2)}$ is the rate of sacrifice bits for the privacy amplification. Hence, the achievable rate of secure CAF is the difference between these two rates.

In fact, all the references [7,8,9,10,11,21] for secure CAF addressed only the case when the number of transmitters is two. Only reference [37] addresses secure CAF when the number of transmitters is larger than two. Unfortunately, these existing studies proposed no application for secure CAF except for a secure two-way relay channel with untrusted relays. The remaining part of this paper discusses its further application.

### 2.3. Concrete Expressions for Mutual Information

In this paper, we employ mutual information $I{\left(Y;A_{1},A_{2}\right)}_{\mathrm{Equation}(2)}-I{\left(Y;A_{1}⊕A_{2}\right)}_{\mathrm{Equation}(2)}$ when $h_{1}=h_{2}=h$. Although their concrete descriptions were presented in ([21], Section IV-A), we give these concrete descriptions here. Assume that $\phi_{a}$ is the Gaussian distribution with average *a* and a variance of one. By using the differential entropy *H*, the mutual information $I{\left(Y;A_{1}⊕A_{2}\right)}_{\mathrm{Equation}(2)}$ is calculated as:(3)$$\begin{array}{c} H\left(\frac{\phi_{0}+2\phi_{h}+\phi_{2h}}{4}\right)-\frac{1}{2}H\left(\frac{\phi_{0}+\phi_{2h}}{2}\right)-\frac{1}{2}H\left(\phi_{h}\right). \end{array}$$
when n→∞, this value goes to log2.

In addition, the mutual information $I{\left(Y;A_{1},A_{2}\right)}_{\mathrm{Equation}(2)}$ is calculated as:(4)$$\begin{array}{c} H\left(\frac{\phi_{0}+2\phi_{h}+\phi_{2h}}{4}\right)-H\left(\phi_{h}\right). \end{array}$$
when n→∞, this value goes to $\frac{3}{2}\log2$.

## 3. Butterfly Network

### 3.1. Conventional Protocol

A typical method for NC is the butterfly NC [22], which efficiently transmits information in the crossing way as explained in Figure 2. The goal of this problem setting is composed of the following two requirements: One is the reliable transmission of the message $M_{1}$ from $V_{1}$ to $V_{6}$, and the other is the reliable transmission of the message $M_{2}$ from $V_{2}$ to $V_{5}$. When each channel transmits only one element of $F_{q}$, the bottleneck of this network is the channel $e_{3}$ from $V_{3}$ to $V_{4}$. Here, no signal is transmitted between disconnected nodes. Hence, no cross talk occurs between disconnected nodes. However, cross talk occurs between $e_{5}$ and $e_{6}$ if the signal on $e_{5}$ is different from that on $e_{6}$. Hence, if they are different, the transmission on $e_{5}$ has to be performed on a different time from the transmission on $e_{6}$. However, when they are the same, these transmissions can be performed simultaneously. In this network model, only the node $V_{3}$ has a freedom to choose the transmitted information because other nodes receive only one information so that they have no other choice for the transmitted information except for transmitting the received information. To resolve the bottleneck in $e_{3}$, the node $V_{3}$ transmits the modulo sum to the node $V_{4}$ via channel $e_{3}$. Then, both destination nodes can recover their respective intended messages while the information transmission over $e_{3}$ is performed only once. That is, the destination node $V_{5}$ decodes the message $M_{2}$ from the received information $M_{1}$ and $M_{1}⊕M_{2}$. Similarly, the other destination node $V_{6}$ decodes the message $M_{1}$ from the received information $M_{2}$ and $M_{1}⊕M_{2}$.

### 3.2. Secure NC

Under the network code given in Section 3.1, the node $V_{3}$ obtains both messages $M_{1}$ and $M_{2}$. The destination node $V_{5}$ recovers the unintended message $M_{1}$ as well as the intended message $M_{2}$, and the other destination node $V_{6}$ the unintended message $M_{2}$ as well as the intended message $M_{1}$. Next, we impose the secrecy against an attack to one of the intermediate (untrusted) nodes. In other words, the information of all intermediate (untrusted) nodes are required to be independent of $M_{1}$ and $M_{2}$, and the information of destination node $V_{5}$ ($V_{6}$) is required to be independent of the unintended message $M_{1}$ ($M_{2}$). This kind of secrecy can be realized under the following assumption. When messages $M_{1}$ and $M_{2}$ are elements of $F_{q}$ and *q* is not a power of 2 in the following way ([38], Figure 2):**(A1)** Two source nodes $V_{1}$ and $V_{2}$ share a secret number *L*,
when the information $Z_{i}$ transmitted on the edge $e_{i}$ is given as:(5)$$\begin{array}{cc} Z_{1} & =2M_{1}⊕L,\;Z_{4}=-\left(M_{1}⊕L\right), \end{array}$$(6)$$\begin{array}{cc} Z_{2} & =2M_{2}⊕L,\;Z_{7}=-\left(M_{2}⊕L\right), \end{array}$$(7)$$\begin{array}{cc} Z_{3} & =Z_{1}⊕Z_{2}=2M_{1}⊕2M_{2}⊕2L, \end{array}$$(8)$$\begin{array}{cc} Z_{5} & =Z_{6}=Z_{3}/2, \end{array}$$(9)$$\begin{array}{cc} {\hat{M}}_{2} & =Z_{5}⊕Z_{4}=M_{2},\;{\hat{M}}_{1}=Z_{6}⊕Z_{7}=M_{1}, \end{array}$$
where ${\hat{M}}_{2}$ (${\hat{M}}_{1}$) is the recovered message by $V_{5}$ ($V_{6}$). Any intermediate edge and any intermediate node obtain no information about the messages $M_{1}$ and $M_{2}$. In addition, the destination node $V_{5}$ ($V_{6}$) obtains no information for the message $M_{1}$ ($M_{2}$) while it obtains the message $M_{2}$ ($M_{1}$). Hence, this code guarantees the following types of security:**(S1)** When the eavesdropper attacks only one of the edges, she obtains no information for each message $M_{i}$.**(S2)** When the nodes do not collude, each node obtains no information for the unintended messages.
When q≥4 is a power of 2, the above code can be modified as follows. We choose an element $e\in F_{q}$ such that $e^{2}⊕e\neq 0$, i.e., e≠1,0.

Then, we define our code as:(10)$$\begin{array}{cc} Z_{1} & =\left(1⊕e\right)M_{1}⊕L,\;Z_{4}=-\left(M_{1}⊕L\right), \end{array}$$(11)$$\begin{array}{cc} Z_{2} & =\left(1⊕e\right)M_{2}⊕eL,\;Z_{7}=-\left(M_{2}⊕L\right), \end{array}$$(12)$$\begin{array}{cc} Z_{3} & =Z_{1}⊕Z_{2}=\left(1⊕e\right)\left(M_{1}⊕M_{2}⊕L\right), \end{array}$$(13)$$\begin{array}{cc} Z_{5} & =Z_{6}=Z_{3}/\left(1⊕e\right), \end{array}$$(14)$$\begin{array}{cc} {\hat{M}}_{2} & =Z_{5}⊕Z_{4}=M_{2},\;{\hat{M}}_{1}=Z_{6}⊕Z_{7}=M_{1}. \end{array}$$

This modification realizes the required security in this case.

### 3.3. Secure PLNC

#### 3.3.1. Use of Secure CAF

If no shared secret number is assumed between $V_{1}$ and $V_{2}$, it is difficult to realize the type of secrecy for the butterfly network presented in Section 3.2 under the problem setting of secure NC. Then, we consider the following assumption:**(A2)** The pairs $\left(e_{1},e_{2}\right)$, $\left(e_{4},e_{5}\right)$, and $\left(e_{6},e_{7}\right)$ are given as Gaussian MACs such as (Equation 2).

In the network model given in Figure 2, only the channel $e_{3}$ is a Gaussian channel with a single input. To achieve secrecy under the assumption (A2), we employ secure CAF in the Gaussian MACs appearing in this network model in the following way: In the Gaussian MAC $\left(e_{1},e_{2}\right)$ at $V_{3}$, the node $V_{3}$ receives only the information $M_{1}⊕M_{2}$. Then, the node $V_{3}$ forwards the received information to the node $V_{4}$, and the node $V_{4}$ receives the information $M_{4}:=M_{1}⊕M_{2}$. In the Gaussian MAC $\left(e_{4},e_{5}\right)$ at $V_{5}$, the node $V_{5}$ receives only the information $M_{4}⊕\left(-M_{1}\right)=M_{2}$. In the same way, the node $V_{6}$ receives only the information $M_{4}⊕\left(-M_{2}\right)=M_{1}$. That is, we employ secure CAF in the three Gaussian MACs at $V_{3}$, $V_{5}$, and $V_{6}$. In this way, these uses of secure CAF realize the security (S2) under this method.

#### 3.3.2. Use of CAF

As another kind of secure PLNC, we attach the CAF to the decoding operations on nodes $V_{3}$, $V_{5}$, and $V_{6}$ in the protocol with q=4 given in Section 3.2. In this protocol, an element of $F_{4}$ is regarded as a vector over the finite field $F_{2}$. While this protocol saves the time, it still requires the secure shared randomness *L*. This protocol can be regarded as a simple combination of secure NC and PLNC.

The assumptions and the realized types of security are summarized in Table 1. Only the protocol given in Section 3.3.1 can realize security (S2) without requiring a secure shared randomness between two source nodes. This is a big advantage for secure PLNC.

### 3.4. Comparison

To implement the above discussed protocols as wireless communication networks, we compare the transmission rates of the protocols given in Section 3.2 and Section 3.3 when each edge is given as the BPSK scheme of a two-input Gaussian channel as (Equation 2) or a single-input Gaussian channel:(15)$$\begin{array}{c} Y=hX+N, \end{array}$$
where h∈C are the channel fading coefficients, *N* is a complex Gaussian random variable with zero mean and a variance of one, and *X* is coded as ${\left(-1\right)}^{A}$ with $A\in F_{2}$. Hence, (Equation 16) is rewritten as:(16)$$\begin{array}{c} Y=h{\left(-1\right)}^{A}+N, \end{array}$$

In this comparison, for simplicity, we assume that $h_{1}=h_{2}=h$. We assume that *T* is the time period of transmitting one Gaussian signal on each edge. Additionally, we assume that ideal codes are available as follows. The mutual information rate $I{\left(Y;A\right)}_{\mathrm{Equation}(16)}$ is achievable over the channel (Equation 16), the rate $I{\left(Y;A_{1}⊕A_{2}\right)}_{\mathrm{Equation}(2)}$ is achievable for CAF in the channel (Equation 2), and the rate $2I{\left(Y;A_{1}⊕A_{2}\right)}_{\mathrm{Equation}(2)}-I{\left(Y;A_{1},A_{2}\right)}_{\mathrm{Equation}(2)}$ is available for secure CAF in the channel (Equation 2). Notice that the relation $I{\left(Y;A_{2}|A_{1}\right)}_{\mathrm{Equation}(2)}=I{\left(Y;A_{1}|A_{2}\right)}_{\mathrm{Equation}(2)}$ holds in this case. In addition, the mutual information rate pair $\left(I{\left(Y;A_{1}A_{2}\right)}_{\mathrm{Equation}(2)}/2,I{\left(Y;A_{1}A_{2}\right)}_{\mathrm{Equation}(2)}/2\right)$ is available in the MAC channel (Equation 2) when both transmitters intend to send their own message to the receiver. (Generally, the symmetric rate $\left(I\left(Y;A_{1},A_{2}\right)/2,I\left(Y;A_{1},A_{2}\right)/2\right)$ is achievable when the symmetric rate $\left(I\left(Y;A_{1},A_{2}\right)/2,I\left(Y;A_{1},A_{2}\right)/2\right)$ belongs to the interval between $\left(I\left(Y;A_{1}\right),I\left(Y;A_{2}|A_{2}\right)\right)$ and $\left(I\left(Y;A_{1}|A_{2}\right),I\left(Y;A_{2}\right)\right)$. Our case with $h_{1}=h_{2}=h$ satisfies this condition.) In the above discussion, the random variables $A_{1}$, $A_{2}$, and *A* are subject to the uniform distribution independently.

The secure NC protocol given in Section 3.2 needs to avoid a crossed line when Gaussian MAC is not used. Now, we consider how much time is needed for this protocol. In this protocol, we need to repeat several processes, each of which is composed of the encoding, wireless communication, and decoding. In the protocol given in Section 3.2, the first step can make the simultaneous transmissions on $e_{1}$ and $e_{4}$. However, the simultaneous transmission on $e_{1}$ cannot be performed simultaneously in order to avoid the cross line on the receiving on $V_{3}$. Hence, the second step makes the simultaneous transmissions on $e_{2}$ and $e_{7}$. We say that the time period for the first step is the time slot of Time i, and the time period for the second step is the time slot of Time ii. That is, each time span for the process composed of the encoding, wireless communication, and decoding is called a time slot. Now, to evaluate the required number of time slots, we assume that all players have only one transmitting antenna, which can broadcast the transmitting signal. Then, we find that the whole network has five time slots at least as presented in Table 2. When the length of the transmitted message is *G*, the transfer time for each time slot is $\frac{GT}{I{\left(Y;A\right)}_{\mathrm{Equation}(16)}}$. Therefore, the total transfer time in this case is calculated to be $\frac{5GT}{I{\left(Y;A\right)}_{\mathrm{Equation}(16)}}$.

When we use the Gaussian MAC, the secure NC protocol given in Section 3.2 can be implemented with three time slots as Table 3 because $V_{4}$ broadcasts the information to $e_{5}$ and $e_{6}$. When the length of the transmitted message is *G*, the first time slot requires transfer time $\frac{2GT}{I{\left(Y;A_{1},A_{2}\right)}_{\mathrm{Equation}(2)}}$, and the second and third time slots require transfer time $\frac{GT}{I{\left(Y;A\right)}_{\mathrm{Equation}(16)}}$. Hence, the total transfer time is calculated to be $\frac{2GT}{I{\left(Y;A\right)}_{\mathrm{Equation}(16)}}+\frac{2GT}{I{\left(Y;A_{1},A_{2}\right)}_{\mathrm{Equation}(2)}}$. When we design the whole process as in Table 4, the first and third time slots require transfer time $\frac{2GT}{I{\left(Y;A_{1},A_{2}\right)}_{\mathrm{Equation}(2)}}$, and the second time slot requires transfer time $\frac{GT}{I{\left(Y;A\right)}_{\mathrm{Equation}(16)}}$. Hence, the total transfer time is $\frac{GT}{I{\left(Y;A\right)}_{\mathrm{Equation}(16)}}+\frac{4GT}{I{\left(Y;A_{1},A_{2}\right)}_{\mathrm{Equation}(2)}}$, which is larger than $\frac{2GT}{I{\left(Y;A\right)}_{\mathrm{Equation}(16)}}+\frac{2GT}{I{\left(Y;A_{1},A_{2}\right)}_{\mathrm{Equation}(2)}}$ because $\frac{I{\left(Y;A_{1},A_{2}\right)}_{\mathrm{Equation}(2)}}{2}\leq I{\left(Y;A\right)}_{\mathrm{Equation}(16)}$.

The secure PLNC protocol given in Section 3.3.1 can be performed only with three time slots as in Table 4, where the pairs $\left(e_{1},e_{2}\right)$, $\left(e_{4},e_{5}\right)$, and $\left(e_{6},e_{7}\right)$ are realized by secure CAF based on the Gaussian MAC channel (Equation 2). The first and third time slots require transfer time $\frac{GT}{2I{\left(Y;A_{1}⊕A_{2}\right)}_{\mathrm{Equation}(2)}-I{\left(Y;A_{1},A_{2}\right)}_{\mathrm{Equation}(2)}}$, and the second time slot requires transfer time $\frac{GT}{I{\left(Y;A\right)}_{\mathrm{Equation}(16)}}$. The total transfer time is calculated to be $\frac{2GT}{2I{\left(Y;A_{1}⊕A_{2}\right)}_{\mathrm{Equation}(2)}-I{\left(Y;A_{1},A_{2}\right)}_{\mathrm{Equation}(2)}}+\frac{GT}{I{\left(Y;A\right)}_{\mathrm{Equation}(16)}}$.

Secure PLNC protocol given in Section 3.3.2 can also be implemented only with three time slots as in Table 4. The first and third time slots require transfer time $\frac{GT}{I{\left(Y;A_{1}⊕A_{2}\right)}_{\mathrm{Equation}(2)}}$, and the second time slot requires transfer time $\frac{GT}{I{\left(Y;A\right)}_{\mathrm{Equation}(16)}}$. The total transfer time is calculated to be $\frac{2GT}{I{\left(Y;A_{1}⊕A_{2}\right)}_{\mathrm{Equation}(2)}}+\frac{GT}{I{\left(Y;A\right)}_{\mathrm{Equation}(16)}}$.

Figure 3 gives the numerical comparison among the following time periods:(17)$$\begin{array}{c} \begin{array}{cc} \; & \frac{2GT}{2I{\left(Y;A_{1}⊕A_{2}\right)}_{\mathrm{Equation}(2)}-I{\left(Y;A_{1},A_{2}\right)}_{\mathrm{Equation}(2)}}+\frac{GT}{I{\left(Y;A\right)}_{\mathrm{Equation}(16)}},\\ \; & \frac{4GT}{I{\left(Y;A\right)}_{\mathrm{Equation}(16)}},\;\frac{2GT}{I{\left(Y;A\right)}_{\mathrm{Equation}(16)}}+\frac{2GT}{I{\left(Y;A_{1},A_{2}\right)}_{\mathrm{Equation}(2)}},\\ \; & \frac{2GT}{I{\left(Y;A_{1}⊕A_{2}\right)}_{\mathrm{Equation}(2)}}+\frac{GT}{I{\left(Y;A\right)}_{\mathrm{Equation}(16)}}. \end{array} \end{array}$$
When h→∞, these values converge to:(18)$$\begin{array}{cc} \; & \frac{5GT}{\log2},\;\frac{4GT}{\log2},\;\frac{10GT}{3\log2},\;\frac{3GT}{\log2}, \end{array}$$
respectively.

The secure NC protocol given in Section 3.2 requires a shorter transfer time for the transmission than the secure PLNC protocol given in Section 3.3 in this comparison. Since the difference is not so extensive, the secure PLNC protocol given in Section 3.3.1 is useful when it is not easy to prepare secure shared randomness between two source nodes. In fact, when the direct communication between two distinct source nodes is not available, we often use the butterfly network. In this case, such a secure shared randomness requires an additional cost. However, the secure PLNC protocol given in Section 3.3.2 has no advantage over the secure NC protocol with the MAC channel. That is, a simple combination of secure NC and PLNC is not useful in this case.

## 4. Network with Three Source Nodes

Finally, we study the network topology shown in Figure 4 that is composed of three source nodes, $S_{1}$, $S_{2}$, and $S_{3}$; three intermediate nodes, $I_{1}$, $I_{2}$, and $I_{3}$; and one destination node, *D*. Its generalization was discussed as a multilayer network in the recent reference [23]. The goal of this network model is the secure transmission from the three source nodes to the destination node *D* when the source node $S_{i}$ is required to send an element $M_{i}\in F_{q}$ to the destination node *D*.

### 4.1. Secure NC

As the first step, let us study the network with three sources under the framework of the secure NC. In Figure 4, every edge expresses a noiseless channel to transmit one element of $F_{q}$. Here, we consider the following two security requirements:**(S3)** When Eve eavesdrops only one edge among three edges (channels) between the intermediate nodes and the destination node, she obtains no information about each message.**(S4)** When Eve eavesdrops only one intermediate (untrusted) node among three intermediate (untrusted) nodes, she obtains no information for each message. Here, no node colludes with another node.

#### 4.1.1. Security (S3)

The following code satisfies Security (S3) when *q* is not a power of 2. This code uses 1/2, which cannot be allowed in finite field $F_{q}$ of a power *q* of 2. Notice that the matrix $\left(\begin{array}{ccc} 0 & 1 & 1\\ 1 & 0 & 1\\ 1 & 1 & 0 \end{array}\right)$ is invertible because $\left(\begin{array}{ccc} -1/2 & 1/2 & 1/2\\ 1/2 & -1/2 & 1/2\\ 1/2 & 1/2 & -1/2 \end{array}\right)$ is the inverse matrix. Source node $S_{i}$ sends $M_{i}$ in each edge. Each intermediate node sends the sum of the received vector. Finally, applying the inverse matrix $\left(\begin{array}{ccc} -1/2 & 1/2 & 1/2\\ 1/2 & -1/2 & 1/2\\ 1/2 & 1/2 & -1/2 \end{array}\right)$ to the received vector, the node *D* recovers all messages. In this code, each of the messages $M_{1}⊕M_{2}$, $M_{2}⊕M_{3}$, and $M_{3}⊕M_{1}$ are independent of anyone of $M_{1}$, $M_{2}$, and $M_{3}$. Hence, Security (S3) is satisfied. This protocol achieves the optimum transmission rate even when the secrecy condition is not imposed.

As the next step, let us proceed to the case when q≥4 is a power of 2. We choose an element $e\in F_{q}$ such that $e^{2}⊕e\neq 0$, which implies that $\left(\begin{array}{ccc} 0 & 1 & 1\\ 1 & 0 & e\\ e & e & 0 \end{array}\right)$ is invertible because its determinant is $e^{2}⊕e\neq 0$. For example, when q=4, since $e^{2}=e⊕1$, the inverse matrix is: $\left(\begin{array}{ccc} e⊕1 & e & e\\ e⊕1 & e & 1\\ e & e & 1 \end{array}\right)$. Then, the following code is secure. Source node $S_{i}$ sends $M_{i}$ in each edge. The intermediate nodes $I_{1}$, $I_{2}$, and $I_{3}$ send the received information $Z_{1}:=M_{2}⊕M_{3}$, $Z_{2}:=M_{1}⊕eM_{3}$, and $Z_{3}:=eM_{1}⊕eM_{2}$, respectively. Finally, applying the inverse matrix of $\left(\begin{array}{ccc} 0 & 1 & 1\\ 1 & 0 & e\\ e & e & 0 \end{array}\right)$ to the received vector $\left(\begin{array}{c} Z_{1}\\ Z_{2}\\ Z_{3} \end{array}\right)$, the node *D* recovers all messages. In this code, each of the information symbols $eM_{1}⊕eM_{2}$, $M_{2}⊕M_{3}$, and $eM_{3}⊕M_{1}$ are independent of anyone of $M_{1}$, $M_{2}$, and $M_{3}$. Hence, Security (S3) is satisfied.

#### 4.1.2. Security (S4)

To make a code satisfy Security (S4), we modify the above protocol as follows. The modified protocol uses the channels between the intermediate (untrusted) nodes and the destination node twice. In addition, it employs the channels between the source nodes and the intermediate (untrusted) nodes only once. Source node $S_{i}$ sends the scrambled variable $L_{i}$ to the intermediate (untrusted) node $I_{i⊕1}$ via the edge $e_{i}$. Each source node $S_{i}$ prepares the scrambled variable $L_{i}$ and sends the variable $M_{i}⊕\left(-L_{i}\right)$ to the intermediate (untrusted) node $I_{i⊕(-1)}$ via the edge $e_{3⊕i}$. Here, i⊕1 and i⊕(−1) are regarded as elements of $Z_{3}$. Each intermediate (untrusted) node sends both received variables to the destination node by using the channel twice. Then, the destination node *D* can recover the messages as:(19)$$\begin{array}{cc} M_{1} & =\left(M_{1}⊕\left(-L_{1}\right)\right)⊕L_{1},\;M_{2}=\left(M_{2}⊕\left(-L_{2}\right)\right)⊕L_{2}, \end{array}$$(20)$$\begin{array}{cc} M_{3} & =\left(M_{3}⊕\left(-L_{3}\right)\right)⊕L_{3} \end{array}$$
because the node *D* obtains information $L_{1}$, $L_{2}$, $L_{3}$, $M_{1}⊕\left(-L_{1}\right)$, $M_{2}⊕\left(-L_{2}\right)$, and $M_{3}⊕\left(-L_{3}\right)$. The information at the intermediate (untrusted) node $I_{i}$ is the pair of $L_{i+1}$ and $M_{i-1}⊕\left(-L_{i-1}\right)$, which is independent of anyone of $M_{1}$, $M_{2}$, and $M_{3}$. Hence, this code guarantees Security (S4) as well as Security (S3).

### 4.2. Secure PLNC

#### 4.2.1. Use of Secure CAF

Now, we assume the following assumption:**(A3)** The channels over the pairs $\left(e_{1},e_{6}\right)$, $\left(e_{2},e_{4}\right)$, and $\left(e_{3},e_{5}\right)$ are Gaussian MACs as in (Equation 2).

That is, the eavesdropper is supposed to access only one of the information symbols at the intermediate (untrusted) nodes, which corresponds to Case 2 of Section 4.1. Then, using secure CAF [21], we construct our protocol.

As the first step, we discuss the case when *q* is not a power of 2. In the Gaussian MAC $\left(e_{1},e_{6}\right)$, we employ secure CAF so that the node $I_{2}$ obtains the information symbol $M_{1}⊕M_{3}$. Similarly, $I_{1}$ and $I_{3}$ obtain the information symbol $M_{2}⊕M_{3}$ and $M_{1}⊕M_{2}$, respectively. Hence, the information symbols at every intermediate (untrusted) node are independent of the messages $M_{1}$, $M_{2}$, and $M_{3}$. In the next step, the intermediates (untrusted) nodes $I_{1}$, $I_{2}$, and $I_{3}$ transmit their received information symbols $M_{1}^{\prime },M_{2}^{\prime },$ and $M_{3}^{\prime }$ to the destination node *D* via the Gaussian MACs with three input signals. Then, applying separate decoding, the destination node *D* recovers the information symbols $M_{1}^{\prime },M_{2}^{\prime },$ and $M_{3}^{\prime }$. Using the method presented in Section 4.1.1, the destination node *D* obtains the original information symbols $M_{1},M_{2},$ and $M_{3}$.

When q≥4 is a power of 2, to apply the method given in Section 4.1.1, the node $I_{2}$ needs to obtain the information $M_{1}⊕eM_{3}$. This task for $I_{2}$ can be implemented by a secure CAF with a two-dimensional vector over the finite field $F_{2}$ by the prior conversion from $M_{3}$ to $eM_{3}$ at the node $S_{3}$ before use of the Gaussian MAC $\left(e_{1},e_{6}\right)$. The same method is applied to the Gaussian MACs $\left(e_{2},e_{4}\right)$ and $\left(e_{3},e_{5}\right)$. The remaining part of this protocol can be performed in the same way as the above.

In the above way, the framework of the secure PLNC enables us to implement the secure code for an attack on an intermediate (untrusted) node by using secure CAF. That is, this code guarantees Security (S4). This protocol requires no additional random variable, unlike the protocol presented in Section 4.1.2.

#### 4.2.2. Use of CAF

Next, we construct a protocol using CAF. In this protocol, at the node *D*, to recover $M_{1}$, we employ CAF on the two edges $e_{8}$ and $e_{9}$. Similarly, to recover $M_{2}$ ($M_{3}$), we employ CAF on the two edges $e_{7}$ and $e_{9}$ ($e_{7}$ and $e_{8}$). To avoid information leakage over every intermediate (untrusted) node, the transmitter applies the secure network code given in Section 4.1.2.

### 4.3. Comparison

All the proposed protocols are summarized in Table 5. Since the security of our interest is (S4), we compare the protocols except for the protocol given in Section 4.1.1. Only the protocol given in Section 4.2.1 satisfies Security (S4). To implement these protocols as wireless communication network, we compare the transmission rates of the protocols given in Section 4.1 and Section 4.2 when each edge is given as the BPSK scheme of a single-input Gaussian channel (Equation 16), a two-input Gaussian channel (Equation 2), or a three-input Gaussian channel (Equation 2):(21)$$\begin{array}{c} Y=hX_{1}+hX_{2}+hX_{3}+N, \end{array}$$
where h∈C are the channel fading coefficients, *N* is a complex Gaussian random variable with zero mean and a variance of one, and $X_{i}$ is coded as ${\left(-1\right)}^{A_{i}}$ with $A_{i}\in F_{2}$. In this comparison, we make the same assumptions for $h_{1}$, $h_{2}$, and GT as the previous section. Additionally, we assume that ideal codes given in Section 3.4 are available, and that the mutual information rate triple
$$\left(\frac{I{\left(Y;A_{1}A_{2}A_{3}\right)}_{\mathrm{Equation}(21)}}{3},\frac{I{\left(Y;A_{1}A_{2}A_{3}\right)}_{\mathrm{Equation}(21)}}{3},\frac{I{\left(Y;A_{1}A_{2}A_{3}\right)}_{\mathrm{Equation}(21)}}{3}\right)$$
is available in the MAC channel (Equation 21) when three transmitters intend to send their own message to the receiver, where the random variables $A_{1}$, $A_{2}$, and $A_{3}$ are independently subject to the uniform distribution [37]. Using this rate, we compare the secure NC protocol given in Section 4.1.2 and the secure PLNC protocol given in Section 4.2 because both protocols realize the secrecy for intermediate (untrusted) nodes.

When any Gaussian MAC is not used, the secure NC protocol given in Section 4.1.2 requires five time slots at least as shown in Table 6. In particular, the edges $e_{7}$, $e_{8}$, and $e_{9}$ need to send the information symbols twice as the remaining edges. Therefore, when the length of the transmitted message is *G*, the first and second time slots need transfer time $\frac{GT}{I{\left(Y;A\right)}_{\mathrm{Equation}(16)}}$, and the remaining time slots need transfer time $\frac{2GT}{I{\left(Y;A\right)}_{\mathrm{Equation}(16)}}$. Hence, the total transfer time is calculated to be $\frac{8GT}{I{\left(Y;A\right)}_{\mathrm{Equation}(16)}}$.

When we use the Gaussian MAC, the secure NC protocol given in Section 4.1.2 can be implemented with two time slots as in Table 7. The first time slot needs transfer time $\frac{2GT}{I{\left(Y;A_{1},A_{2}\right)}_{\mathrm{Equation}(2)}}$, and the second time slot needs transfer time $\frac{6GT}{I{\left(Y;A_{1}A_{2}A_{3}\right)}_{\mathrm{Equation}(21)}}$. Hence, the total transfer time is calculated to be $\frac{6GT}{I{\left(Y;A_{1}A_{2}A_{3}\right)}_{\mathrm{Equation}(21)}}+\frac{2GT}{I{\left(Y;A_{1},A_{2}\right)}_{\mathrm{Equation}(2)}}$.

The secure PLNC protocol given in Section 4.2.1 can be implemented only with two time slots as in Table 8, where the pairs $\left(e_{1},e_{2}\right)$, $\left(e_{4},e_{5}\right)$, and $\left(e_{6},e_{7}\right)$ are realized by the secure CAF based on the Gaussian MAC channel (Equation 2). The first time slot needs transfer time $\frac{GT}{2I{\left(Y;A_{1}⊕A_{2}\right)}_{\mathrm{Equation}(2)}-I{\left(Y;A_{1},A_{2}\right)}_{\mathrm{Equation}(2)}}$, and the second time slot needs transfer time $\frac{3GT}{I{\left(Y;A_{1}A_{2}A_{3}\right)}_{\mathrm{Equation}(21)}}$. Hence, the total transfer time is calculated to be $\frac{3GT}{I{\left(Y;A_{1}A_{2}A_{3}\right)}_{\mathrm{Equation}(21)}}+\frac{GT}{2I{\left(Y;A_{1}⊕A_{2}\right)}_{\mathrm{Equation}(2)}-I{\left(Y;A_{1},A_{2}\right)}_{\mathrm{Equation}(2)}}$.

Another secure PLNC protocol given in Section 4.2.2 can be implemented only with two time slots as in Table 9, where the pairs $\left(e_{1},e_{2}\right)$, $\left(e_{4},e_{5}\right)$, and $\left(e_{6},e_{7}\right)$ are realized by the secure CAF based on the Gaussian MAC channel (Equation 2). The first time slot needs transfer time $\frac{2GT}{I{\left(Y;A_{1},A_{2}\right)}_{\mathrm{Equation}(2)}}$, and the second time slot needs transfer time $\frac{3GT}{I{\left(Y;A_{1}⊕A_{2}\right)}_{\mathrm{Equation}(2)}}$. Hence, the total transfer time is calculated to be $\frac{3GT}{I{\left(Y;A_{1}⊕A_{2}\right)}_{\mathrm{Equation}(2)}}+\frac{2GT}{I{\left(Y;A_{1},A_{2}\right)}_{\mathrm{Equation}(2)}}$.

Figure 5 gives the numerical comparison among the following time periods:(22)$$\begin{array}{c} \begin{array}{cc} \; & \frac{8GT}{I{\left(Y;A\right)}_{\mathrm{Equation}(16)}},\;\frac{6GT}{I{\left(Y;A_{1}A_{2}A_{3}\right)}_{\mathrm{Equation}(21)}}+\frac{2GT}{I{\left(Y;A_{1},A_{2}\right)}_{\mathrm{Equation}(2)}},\\ \; & \frac{3GT}{I{\left(Y;A_{1}⊕A_{2}\right)}_{\mathrm{Equation}(2)}}+\frac{2GT}{I{\left(Y;A_{1},A_{2}\right)}_{\mathrm{Equation}(2)}}.\\ \; & \frac{3GT}{I{\left(Y;A_{1}A_{2}A_{3}\right)}_{\mathrm{Equation}(21)}}+\frac{GT}{2I{\left(Y;A_{1}⊕A_{2}\right)}_{\mathrm{Equation}(2)}-I{\left(Y;A_{1},A_{2}\right)}_{\mathrm{Equation}(2)}}. \end{array} \end{array}$$
When h→∞, these values converge to:(23)$$\begin{array}{c} \frac{8GT}{\log2},\;\frac{8GT}{4\log2-\log3}+\frac{4GT}{3\log2},\;\frac{13GT}{3\log2},\;\frac{4GT}{4\log2-\log3}+\frac{2GT}{\log2}, \end{array}$$
respectively.

The codes for the secure PLNC protocol given in Section 4.2.1 require shorter transfer time for the transmission than the secure NC protocol given in Section 4.1.2 in this comparison when the coefficient *h* is larger than about 1.7. This comparison shows that the secure PLNC protocol given in Section 4.2.1 has an advantage over the secure NC protocol given in Section 4.1.2 when the power of the signal is sufficiently large. In addition, this comparison indicates the advantage of the simple combination of secure NC and PLNC given in Section 4.2.2 over the secure NC protocol given in Section 4.1.2 with the MAC channel. In other words, if the power of the signal is not so large, the secure NC protocol given in Section 4.1.2 with the MAC channel is better than the secure PLNC protocols given in Section 4.2.1 and Section 4.2.2.

## 5. Conclusions and Discussion

We have studied the advantages of a secure PLNC over a secure NC. To investigate this type of advantage, we have focused on two typical network models. Section 3 has discussed the butterfly network model given in Figure 2, and Section 4 has discussed the network model with three source nodes given in Figure 4. We have described concrete protocols that efficiently realize the required secrecy and work over these network models. In these examples, the secure PLNC can realize the secrecy even with untrusted intermediate nodes. In particular, as summarized in Table 1, in the butterfly network, although the protocols using secure network codes require a secure shared randomness for this purpose, the secure PLNC does not need it. Comparing the transfer times of the proposed codes, we have shown that the secure PLNC has a shorter transfer time than the the simple combination of secure NC and physical layer network under a certain range of channel parameters.

As one of the main reasons of these advantages, we can list the fact that secure PLNC is a cross-layered network protocol. That is, it can be realized by a joint application of the error correction and the secure NC by using the mechanism of a physical layer while the conventional scenario can be considered as separate application of the error correction and the secure NC. In particular, the noise in the channels is utilized for keeping the secrecy in the secure PLNC. Therefore, we can conclude that the secure PLNC is useful to realize the secrecy against information leakage at intermediate (untrusted) nodes.

One might consider that the proposed method does not work for jamming attacks [39] or spoofing [40,41]. The transmitters and the receivers can detect it by attaching authentication [42,43,44,45], which can be realized by using a universal2 hash function and preshared keys.

Furthermore, the number of existing applications of the secure PLNC is quite limited. It is an important future study to find much more fruitful applications of the secure PLNC over untrusted relays. In fact, reference [21] also derived an upper bound for the amount of the leaked information of the constructed finite-length code. Therefore, it is an interesting future topic to make finite-length analysis by applying the finite-length analysis in [21]. In addition, the analysis of this paper is based on the BPSK scheme. Since many papers on secure PLNC were based on lattice codes, a similar comparison based on lattice codes is needed. Such a comparison remains an interesting open problem.

Finally, we list three future problems. The first one is the application of the proposed method to multi-hop untrusted relaying networks [8,46]. The second one is the realization of covert communication [47,48] over the wireless networks discussed in this paper. The third one is the problem related to retransmission. In real communication, there is a possibility that we need to perform retransmission due to various reasons. While such a retransmission causes delay, our time analysis does not cover it. In addition, due to the existence of retransmissions, the network needs to prepare a certain central system that controls the status of the whole network. It is another future problem to design the implementation of our system taking care of these issues. These are challenging future studies.

## Figures and Tables

**Figure 1 entropy-24-00047-f001:**
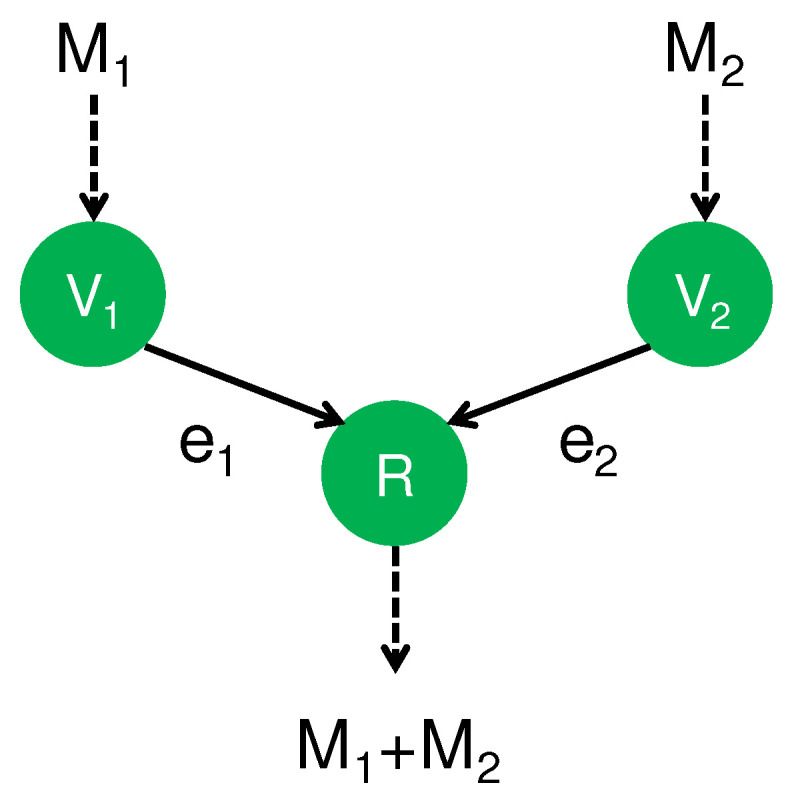
CAF (Computation-and-forward).

**Figure 2 entropy-24-00047-f002:**
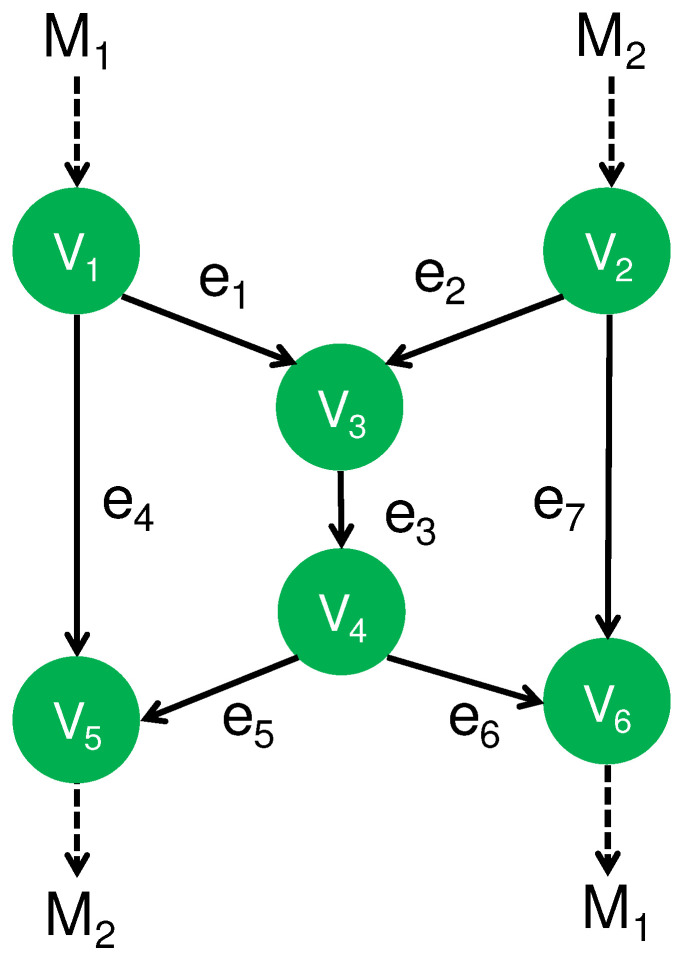
Butterfly NC.

**Figure 3 entropy-24-00047-f003:**
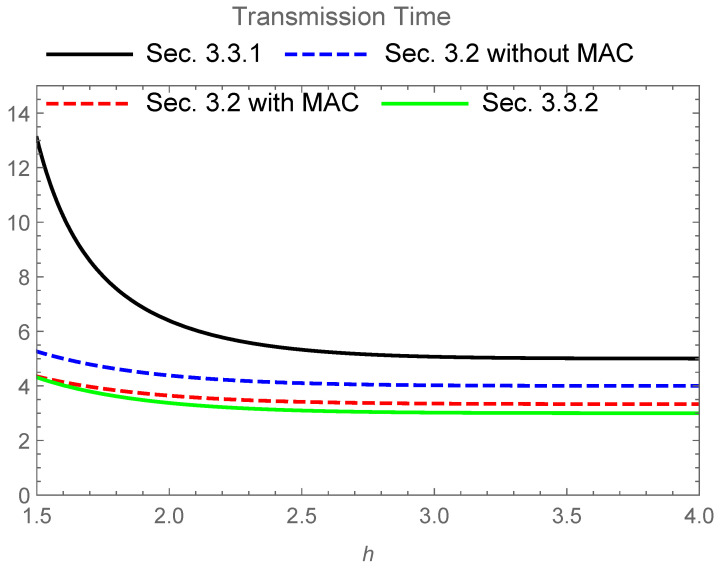
Transmission time for four schemes when GT=1 and the base of the logarithm is 2. The upper solid line (black) expresses the time $\frac{2GT}{2I{\left(Y;A_{1}⊕A_{2}\right)}_{\mathrm{Equation}(2)}-I{\left(Y;A_{1},A_{2}\right)}_{\mathrm{Equation}(2)}}+\frac{GT}{I{\left(Y;A\right)}_{\mathrm{Equation}(16)}}$ of the secure PLNC protocol given in Section 3.3.1. The upper dashed line (blue) expresses the time $\frac{4GT}{I{\left(Y;A\right)}_{\mathrm{Equation}(16)}}$ of the secure NC protocol given in Section 3.2 without the MAC channel. The lower dashed line (red) expresses the time $\frac{2GT}{I{\left(Y;A\right)}_{\mathrm{Equation}(16)}}+\frac{2GT}{I{\left(Y;A_{1},A_{2}\right)}_{\mathrm{Equation}(2)}}$ of the secure NC protocol given in Section 3.2 with the MAC channel. The lower solid line (green) expresses the time $\frac{2GT}{I{\left(Y;A_{1}⊕A_{2}\right)}_{\mathrm{Equation}(2)}}+\frac{GT}{I{\left(Y;A\right)}_{\mathrm{Equation}(16)}}$ of the secure PLNC protocol given in Section 3.3.2.

**Figure 4 entropy-24-00047-f004:**
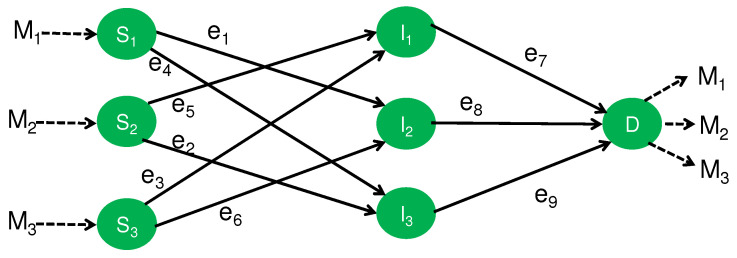
Network with three sources.

**Figure 5 entropy-24-00047-f005:**
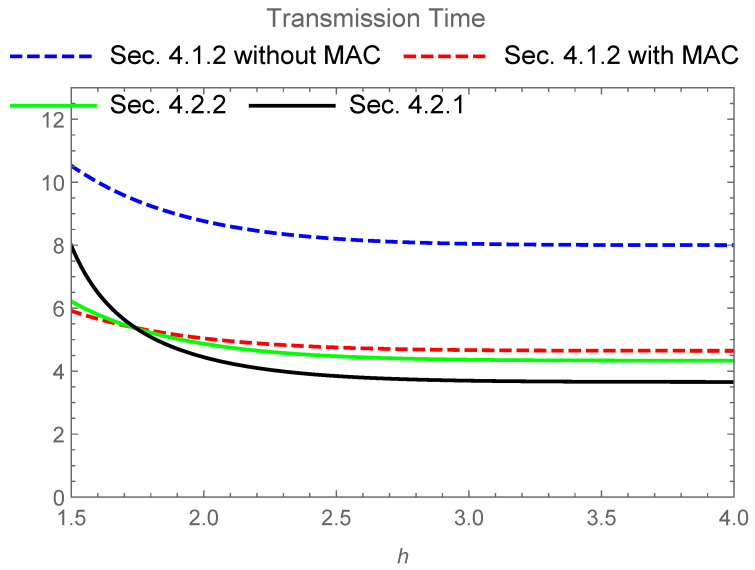
Transmission Time for four schemes when GT=1 and the base of the logarithm is 2. The upper dashed line (blue) expresses the time $\frac{8GT}{I{\left(Y;A\right)}_{\mathrm{Equation}(16)}}$ of secure NC protocol given in Section 4.1.2 without the MAC channel. The lower dashed line (red) expresses the time $\frac{6GT}{I{\left(Y;A_{1}A_{2}A_{3}\right)}_{\mathrm{Equation}(21)}}+\frac{2GT}{I{\left(Y;A_{1},A_{2}\right)}_{\mathrm{Equation}(2)}}$ of the secure NC protocol given in Section 4.1.2 with the MAC channel. The solid line (green) expresses the time $\frac{3GT}{I{\left(Y;A_{1}⊕A_{2}\right)}_{\mathrm{Equation}(2)}}+\frac{2GT}{I{\left(Y;A_{1},A_{2}\right)}_{\mathrm{Equation}(2)}}$ of the secure PLNC protocol given in Section 4.2.2. The solid line (black) expresses the time $\frac{3GT}{I{\left(Y;A_{1}A_{2}A_{3}\right)}_{\mathrm{Equation}(21)}}+\frac{GT}{2I{\left(Y;A_{1}⊕A_{2}\right)}_{\mathrm{Equation}(2)}-I{\left(Y;A_{1},A_{2}\right)}_{\mathrm{Equation}(2)}}$ of the secure PLNC protocol given in Section 4.2.1. The solid line (black), the solid line (green), and the lower dashed line (red) intersect around h=1.7.

**Table 1 entropy-24-00047-t001:** Comparison for protocols in butterfly network.

Protocol	Assumption	Security	Type
Section 3.2	(A1)	(S1) (S2)	Secure NC
Section 3.3.1	(A2)	(S2)	Secure PLNC with secure CAF
Section 3.3.2	(A1) (A2)	(S1) (S2)	Secure PLNC with CAF and secure NC

**Table 2 entropy-24-00047-t002:** Secure NC without Gaussian MAC.

Time Slot	Time i	Time ii	Time iii	Time iv	Time v
Channel	$e_{1}$, $e_{4}$	$e_{2}$, $e_{7}$	$e_{3}$	$e_{5}$	$e_{6}$

**Table 3 entropy-24-00047-t003:** Secure NC with Gaussian MAC.

Time Slot	Time i	Time ii	Time iii
Channel	$\left(e_{1},e_{2}\right)$	$e_{3}$, $e_{4}$, $e_{7}$	$e_{5}$, $e_{6}$

$\left(e_{i},e_{j}\right)$ expresses a Gaussian MAC composed of the joint transmission on the edges $e_{i}$ and $e_{j}$.

**Table 4 entropy-24-00047-t004:** Secure PLNC with Gaussian MAC.

Time Slot	Time i	Time ii	Time iii
Channel	$\left(e_{1},e_{2}\right)$	$e_{3}$	$\left(e_{4},e_{5}\right)$,$\left(e_{6},e_{7}\right)$

**Table 5 entropy-24-00047-t005:** Comparison for protocols in network given in Section 4.1 and Section 4.2.

Protocol	Assumption	Security	Type
Section 4.1.1	–	(S3)	Secure NC
Section 4.1.2	–	(S3) (S4)	Secure NC
Section 4.2.1	(A3)	(S4)	Secure PLNC with secure CAF
Section 4.2.2	(A3)	(S3) (S4)	Secure PLNC with CAF and secure NC

**Table 6 entropy-24-00047-t006:** Secure NC without Gaussian MAC.

Time Span	Time i	Time ii	Time iii	Time iv	Time v
Channel	$e_{1},e_{2},e_{3}$	$e_{4},e_{5},e_{6}$	$e_{7}$	$e_{8}$	$e_{9}$

**Table 7 entropy-24-00047-t007:** Secure NC with Gaussian MAC.

Time Span	Time i	Time ii
Channel	$\left(e_{1},e_{6}\right)$, $\left(e_{2},e_{4}\right)$,$\left(e_{3},e_{5}\right)$	$\left(e_{7},e_{8},e_{9}\right)$

**Table 8 entropy-24-00047-t008:** Secure PLNC with secure CAF.

Time Span	Time i	Time ii
Channel	$\left(e_{1},e_{6}\right)$, $\left(e_{2},e_{4}\right)$, $\left(e_{3},e_{5}\right)$	$\left(e_{7},e_{8},e_{9}\right)$

**Table 9 entropy-24-00047-t009:** Secure PLNC with CAF.

Time Span	Time i	Time ii	Time iii	Time iv
Channel	$\left(e_{1},e_{6}\right)$, $\left(e_{2},e_{4}\right)$,$\left(e_{3},e_{5}\right)$	$\left(e_{8},e_{9}\right)$	$\left(e_{7},e_{9}\right)$	$\left(e_{7},e_{8}\right)$

## Data Availability

Data sharing not applicable.

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
