# Peer review of "Secure Physical Layer Network Coding versus Secure Network Coding†"

_entropy, 2021, doi:10.3390/e24010047_

Round 1

Reviewer 1 Report

The authors of the paper compare the secure physical layer network coding versus the secure network coding. Considering that the channel is complex AWGN, the modulation BPSK and the channel of all paths is the same, then it is concluded that for the butterfly network always the secure physical layer network coding provides a lower transmission time than the secure network coding. On the contrary for the case of a network with three sources, secure physical layer coding provides the lowest transmission time when the channel is larger than h=1.7. Otherwise the secure network coding has the lowest transmission time.  Regarding the security, security physical layer coding, which can be understood as a combination of error correction and secure network coding, helps to improve the secrecy against information leakage at the intermediate nodes.  

After reading the paper my feedback is the following:

1) In the introduction comment the applications in which can be applied secure physical layer network coding. In the butterfly and the network structure of three sources is assumed that the channel is the same. Consequently, specify the type of scenario in which this assumption will be valid. 

2) In Eq. 2 is considered that the BPSK symbols can be rewritten by (-1)^A_1 and (-1)^A_2. All the following notation is expressed in terms of A1 and A2.  Here my concerns are the following.1) What it is the advantage of using this model respect to using the symbols BPSK s1 and s2? , 2) What happen if it is desired to use a higher modulation? 3) What happen if the modulation is complex? I guess that using directly the modulated symbols is easier in terms of notation. 

3) The mutal information also depends on the channel values. So I(Y;A_1 or_exclusive A2) should be : I(Y;A_1 or_exclusive A2;h1;h2). Similar for all values of the mutual information. I understand that the channel is perfectly known, but it should be considered since it is a random variable. Otherwise, the formulation that you are considering is not aligned with the real physics.

4) Page 5, introduce equation numbers in the set of equations of the infomation that it is transmitted on the edges. 

5) Introduce any plot where the variable Z_i be represented. Otherwise, it is lost the idea. 

6) The variable Z from (3) is different from the variable Z_i of page 5. The first one is a complex Gaussian random variable whereas the second one is a BPSK one. 

7) In page 3 uses R for modeling the receiver and in page 7 for specifying the length of the transmitted message. Please, change the name of the variable of both or one of them. No use the same variable for two different concepts. 

8) The values of the mutual informations for each case should be explained with more detail. Some graphic could be helpful. 

9) In Figures 3 and 5 uses the approximation of RT=1. Then, I would refer to the transmission time that you plot as the normalized Transmission time. 

10)  Introduce number of equations in page 8 and explain a bit more how are obtained such values. 

11) The time analysis does not consider the retransmissions due to different channels or a band channel. It should be introduced in the conclusions and the theoretical analysis that it has not been assumed the delay produced by the retransmissions. 

12) The authors comment that secure physical network coding is robust to eavesdropping. However, if there were a man-in-the-middle attack, i.e. the injection of incorrect data to the network, I guess that this architecture would fail. Specially, if the SNR of the man-in-the-middle signal had a larger SNR than the legitimate one. So, the nodes should be equipped with any anti-jamming technique.

13) Given that it can exists retransmissions, the network should have any central system that control the status of the network. Some comment about that should be of interest. 

14) For the second case, the network of three sources, the network physical layer security coding is better than the network security coding when the channel coefficient is larger than h=1.7. The point is that the channels are random variables with a deviation. So, it should be computed the Cumulative Density Probability of the channel for having a larger probablity than 1.7 for difference deviatons. My understanding is that a value of channel of 1.7 may be quite large for certain cases and so always is better to use network secure coding. For that reason is always interesting to introduce the channel in the notation of mutual information. 

15) Papers 27 and 28 provide the same information. Reference 27 may be removed. Introduce references addressed to scenarios in which the assumption that this paper assumes can be introduce for secure the network.  For instance, it is not said anything about if it can be applied to terrestrial, satellite, indoor, etc. This could help to sell the paper.

Reviewer 2 Report

This manuscript is a mostly theoretical comparison between secure network coding and several forms of secure physical network coding in two canonical topologies. The author derives the security properties and the transmission time of each scheme, and determines the advantages for each protocol.

The topic is highly interesting and timely. The manuscript has a number of minor concerns:

1) all the expressions for the rate are based on mutual information terms e.g., I(Y; A_1, A_2). These are actually the Shannon capacities of the point-to-point or multiple-access channels, which have a closed-form expression in terms of the channel fading coefficient h. Maybe it would be more illuminating for the performance results to give their expressions as functions of the SNR.

2) In Section 3.3.1 what are the (B4) and (B3) security?

3) In line 222, why is the total time 4RT/I(Y; A) and not 5RT/I(Y; A) if 5 time slots are used?

4) In the MAC channels, the symmetric rates (I(Y; A1, A2)/2, I(Y; A1, A2)/2) and (I(Y; A1, A2, A3)/3, ..., ...) are possible only because the channel fading coefficients h are equal for all the cases. I believe it is important to highlight this.

5) In Figs 3 and 4, please add a legend to the plots. Also, clearly in the plots we see an asymptotic value when h -> nifty. Is it possible to calculate this limiting value?

6) Paragraph in lines 311 to 316 describes the solution for filed with q >= 4, but actually works for q=4 only, I guess.

Please, revise carefully the above points.

Reviewer 3 Report

The paper gives a nice tutorial on secure physical layer network coding and secure networking coding and the difference between them.  The relative assumptions of the two techniques for privacy and security are also described which are important those who might be selecting between these techniques.

I think that there is good work here but there a few issues that I think should be clarified in the final manuscript.

  1. The paper should be make the novel contribution of this paper versus the cited previous works on these topics.  
  2. There are some minor grammar issues with the paper that should be corrected before the final version is submitted.
  3. The physical layer security method versus the network security methods are very much a function of the communications channel capacity and the function of the channel model, receiver noise, and the channel interference between different communicators.  The capacity formulas I(Y;A1⊕A2) and I(Y;A1,A2) should be expanded in full so people can properly compare these techniques.
  4. It might be nice to provide some further information on the algorithmic complexity of the techniques.  Many of the physical layer security techniques provide good information theoretical bounds on security but the algorithms needed to exploit these relative gains have complexities that would make these algorithms infeasible for many applications.  This would be a useful additional factor to consider when comparing these techniques.

If these problems are corrected, the paper would be suitable for publication.

Round 2

Reviewer 1 Report

After reading the new version of the document I guess that the authors have done the major part of th required changes and correctly justified the recommendations that they have not introduced. For me the paper can be published.